# Antimicrobial Compounds in Food Packaging

**DOI:** 10.3390/ijms24032457

**Published:** 2023-01-27

**Authors:** Aleksandra Duda-Chodak, Tomasz Tarko, Katarzyna Petka-Poniatowska

**Affiliations:** 1Department of Fermentation Technology and Microbiology, Faculty of Food Technology, University of Agriculture in Krakow, ul. Balicka 122, 30-149 Kraków, Poland; 2Department of Plant Products Technology and Nutrition Hygiene, Faculty of Food Technology, University of Agriculture in Krakow, ul. Balicka 122, 30-149 Kraków, Poland

**Keywords:** foodborne pathogens, smart packaging, active packaging, food coatings, antibacterial agents

## Abstract

This review presents current knowledge on antimicrobial agents that are already used in the food packaging industry. At the beginning, innovative ways of food packaging were discussed, including how smart packaging differs from active packaging, and what functions they perform. Next, the focus was on one of the groups of bioactive components that are used in these packaging, namely antimicrobial agents. Among the antimicrobial agents, we selected those that have already been used in packaging and that promise to be used elsewhere, e.g., in the production of antimicrobial biomaterials. Main groups of antimicrobial agents (i.e., metals and metal oxides, organic acids, antimicrobial peptides and bacteriocins, antimicrobial agents of plant origin, enzymes, lactoferrin, chitosan, allyl isothiocyanate, the reuterin system and bacteriophages) that are incorporated or combined with various types of packaging materials to extend the shelf life of food are described. The further development of perspectives and setting of new research directions were also presented.

## 1. Introduction

Food is the source which provides essential nutrients to the human body. Unfortunately, it is also a very good environment for numerous microorganisms to develop in, including those that cause food spoilage. Many of these microorganisms are also human pathogens and pose a real challenge for modern science. Among the most common foodborne pathogens are bacteria (*Escherichia coli, Listeria monocytogenes, Staphylococcus aureus, Bacillus cereus, Campylobacter jejuni, Clostridium botulinum, C. perfringens, Cronobacter sakazakii, Salmonella* spp., *Shigella* spp., *Vibrio* spp., and *Yersinia enterocolitica*), viruses (Hepatitis A and noroviruses) and parasites (*Cyclospora cayetanensis, Toxoplasma gondii* and *Trichinella spiralis*) [1].

Pathogens are ubiquitous and cause a variety of human diseases which are often life-threatening. Along with the development of medical science and progress in health care, pathogen entry the human body as contamination of biomedical products began to be an increasing burden for the system. The biomaterial introduced into the patient’s body is intended to save human life or improve its quality, but instead it becomes a source of infection that potentially threatens health or might be hazardous for life of the patient [2]. Hospital-acquired infections (HAI), which can be caused by bacteria, fungi and viruses, are a significant complication for implantation procedures of various biomaterials, both during planned surgical operations (various prostheses, implants, dentures, tissue scaffolds, stents, heart valves, hip joint endoprostheses), as well as during life-saving actions in the intensive care unit (vascular catheters, urinary catheters, fracture-fixation devices) [3]. The most important pathogens causing biomaterial-assisted infections (BAI) include *Staphylococcus aureus, S. epidermidis, S. saprophyticus, Escherichia coli, Klebsiella pneumoniae, Proteus mirabilis, P. vulgaris, Enterococcus faecalis, Streptococcus viridans, Propionibacterium acnes*, *Pseudomonas aeruginosa*, and *Candida* sp. [3,4]. HAI are amongst the main complications associated with the implantation of any biomaterial. Hence, modern science strives to develop new agents/substances with antibacterial (or, more broadly, biocidal) potential which can be integrated with the biomaterial to prevent bacterial colonization and biofilm formation. These compounds, in addition to a broad spectrum of antimicrobial properties, should be characterized by minimal toxicity to humans. Bacteria can get into the biomaterial in various ways, but usually the source of infection is the direct contamination of the biomaterial with bacteria present in the operating room or with bacteria normally inhabiting the skin (of the patient, doctor or nurse) during the introduction of the device into the subject’s body [4]. Less often, the infection occurs during hospitalization or later (e.g., even several years after surgery), when bacteria originating from a local infection of the patient, via blood, get to the site of biomaterial implantation and cause the development of a new contamination.

In the food production process, the situation is similar, and products can be contaminated at various stages of the food chain. Microorganisms get into the food from the environment and primary production (raw material, production lines, equipment for processing and transporting raw materials, semi-finished and ready-made food), as well as from the skin of personnel who do not properly care about the hygiene of the process. They can be present both in the raw material used for production, as well as being introduced at subsequent stages, during improper storage, transport, distribution of finished products or even during consumption [5,6].

Of course, food spoilage causes economic loss in the food industry and can damage a producer’s reputation, even resulting in a punishment according to the local law. Not all microorganisms that lead to food spoilage are dangerous for human, but the development and the contamination of food with pathogens or their toxins impair the safety of food and pose a risk to the health of consumers [6]. It is estimated that each year 48 million Americans get sick, 128,000 are hospitalized, and 3000 die due to foodborne diseases [7]. The top five pathogens that cause domestically acquired foodborne diseases in USA are noroviruses (58%), followed by non-typhoidal *Salmonella* spp. (11%), *Clostridium perfringens* (10%), *Campylobacter* spp. (9%) and *Staphylococcus aureus* (3%). However, among the causes of hospitalizations and deaths, *Escherichia coli* O157, *Toxoplasma gondii* and *Listeria monocytogenes* are also mentioned [7]. According to WHO, more than 23 million Europeans became ill (4654 of deaths) from eating contaminated food in 2010. The most frequent causes were noroviruses, *Campylobacter* spp., non-typhoidal *Salmonella enterica*, *Toxoplasma gondii*, *Giardia* spp., *Cryptosporidium* spp., Shiga toxin producing *E. coli*, enteropathogenic *E. coli*, as well as hepatitis A virus, *Shigella* spp., *L. monocytogenes, Brucella* sp., *Mycobacterium bovis*, and *Echinococcus* spp. [8]. These diseases generate huge costs related to treatment, absence from work or compensation.

Therefore, considering both safety and economic considerations, manufacturers are trying to develop new food packaging that, in addition to the “traditional” function, will have antimicrobial properties. This review paper aims to gather in one place information on the antimicrobial compounds that are used in the production of food packaging but which have or potentially have a predisposition to be used in the production of antimicrobial biomaterials.

## 2. Food Packaging

The main functions of the food packaging are usually passive, chemical, biological, and physical protection of food against secondary contamination, environmental influences, mechanical damage during storage and transport, as well as against access to light, oxygen, UV radiation, or moisture. However, the packaging also has an informative (name of the product, information on chemical composition and allergens, expiration date, method of preparation and administration, etc.) and promotional (information about the manufacturer) function [9]. In recent years, packaging has also started to play an active role in preventing food waste by actively extending its shelf life while ensuring quality and safety [10].

Traditionally, the main kinds of materials used for food packaging are glass, metals, paper, and plastic, and their main common characteristics were neutrality to the packaged food [9]. What was initially the main advantage of plastic (its chemical resistance, low price and weight) has become a disadvantage, causing these issues to dominate the packaging market and have a huge negative impact on the environment. The main polymers used in food packaging industry are polyethylene (HDPE, LDPE, and LLDPE), polystyrene (PS), polypropylene (PP), polyvinylchloride (PVC), polyamide (PA), and polyethylene terephthalate (PET) [11,12]. To minimize environmental impact, some kinds of food packaging have started to be recycled or even reused (mainly glass and metals), but the recycling of plastic is still insufficient. Therefore, fully biodegradable, safe biopolymers are constantly under development and undergoing a process of improvement [13].

Biopolymers that are gaining popularity in the packaging industry are polysaccharides (e.g., starch and modified starches, cellulose, pectin, carrageenan, alginate, chitin, chitosan), lipids (resins, bee waxes, natural resins, lecithin, emulsifiers, etc.), proteins (casein, collagen, gelatin, gluten, zein, soy proteins, etc.), and microbial polymers (pullulan, curdlan, gellan, xanthan polyhydroxyalkanoates/PHA/, polyhydroxybutyrate/PHB/), polylactic acid/PLA/, etc.) [10,12,13,14]. Although they are all biodegradable and renewable, their “packaging values” are not perfect (e.g., due to weak mechanical properties, hydrophilicity, crystallization, aggregation) and still require improvement. However, many of these biopolymers can be variously modified by methods that improve their barrier function or completely change their characteristics. Thanks to that, bio-based polymers easily found their place in the industry and have different applications. Among the examples of this are the derivatives of cellulose [13] or chitosan [15]. Moreover, new bio-based materials were introduced recently and, in what is exactly the opposite of traditional food packaging, they should interact with the packaged food, albeit in a desirable and planned way. Such innovative packaging systems were created from various biopolymers used as a packaging matrix, to which additional ingredients with an important function were introduced. Among such innovative systems are ‘active’ and ‘smart’ packaging and their characteristics are presented below.

### 2.1. Smart Packaging

“Smart materials and articles” means materials and articles that monitor the condition of packaged food or its environment [Commission Regulation (EC) No 450/2009]. Smart packaging is designed to inform about the product in a more advanced way than traditional packaging. It not only gives knowledge about the composition or the manufacturer, it allows the tracking of the fate of a food product throughout the chain, including monitoring changes that have taken place in the environment and/or in the product (e.g., changes in temperature, pH, acidity). Smart packaging communicates to the consumer whether the product still meets all the requirements, whether it is safe, for example, whether the cold chain has been disturbed during the manufacturing process, storage and transport of the food product, or whether anything else has happened that could adversely affect the safety and quality of that food [16,17]. The new generation of smart packaging is equipped with biosensors (e.g., ToxinGuard^®^; Food Sectinel System^®^). They may communicate if any harmful or undesirable metabolites (such as toxins or biogenic amines) have appeared in food product or if dangerous food pathogens have developed [18,19]. Thus it is an intelligent way to monitor food quality. Furthermore, smart packaging indirectly allows us to extend shelf life. First, it is a useful tool that enhances product traceability, allows to control the shelf life of the product throughout the supply chain, and thus allows to make better decisions regarding the order of transport, as well as in terms of sales policy and product promotion in the store. Secondly, especially in the case of products with a ‘best before’ date, it can prevent a healthy and wholesome product from being thrown away. Consumers, seeing that a product has ‘exceeded’ the date given on the packaging, often throw food away without thinking because they cannot distinguish between the term ‘best before date’ from the term ‘use-by-date’. This leads to a lot of food loss. The stated ‘best before date’ only defines the minimum period for which all product characteristics are guaranteed, while the actual shelf life is longer, although it largely depends on how the product has been stored. Therefore, smart packaging, by informing the consumer about the condition of the product (e.g., whether the pH has dropped too much, whether the product has been refrigerated all the time, whether harmful components are present in the packaging) can prevent food from being thrown away and the waste of something that still could be safely used [19].

### 2.2. Active Packaging

The term “active” means that it has additional functions consisting of actively caring for the quality and safety of food and extending its shelf life, mainly by providing additional protection against the growth of microorganisms or the oxidation of a product. According to Commission Regulation (EC) No 450/2009, “active materials and articles” means materials and articles intended to extend the shelf life or to maintain or improve the condition of packaged food, and they are intentionally designed in such a way that the ingredients they contain release or absorb substances into the packaged food or its surroundings. Consequently, active packaging can act by: (i) managing the atmosphere inside the package (for example, by binding oxygen or emitting CO_2_); (ii) preventing the development of foodborne pathogens and/or by inhibiting or killing food spoilage microorganisms, (iii); inhibiting the formation of biofilm; (iv) exerting anti-radical and antioxidant effects; or (v) by absorbing substances which can cause food spoilage or have an unpleasant odor (ethanol, acids, oxygen, carbon oxide, ammonia) [14,16].

Since microorganisms are one of the main reasons why food loses its quality or ceases to be safe, the greatest interest is aroused by active packaging, whose task is to inhibit the growth of microorganisms or limit the release of harmful metabolites by them (including toxins, gases, acids).

## 3. Antimicrobial Agents and Their Application in Food Industry

Striving to produce healthy, high-quality food while limiting chemical preservatives, combined with the desire to reduce the impact of packaging on the environment, causes scientists and food producers to focus on producing the most natural packaging containing natural antimicrobial agents. It is not possible to find a single antimicrobial agent able to inhibit all microorganisms that cause food spoilage or food poisoning. However, scientists have checked hundreds of potential candidates to find the most potent. Moreover, also some packaging materials can exert antimicrobial activity (e.g., chitosan). Usually, antimicrobial packaging is developed by loading or coating an antimicrobial agent onto polymeric packaging films, and an antimicrobial component can migrate to the food’s surface (migrating film) or can act against surface microbial growth without migration (non-migrating film) [16].

Active agents used in food packaging can be classified according to various criteria, such as origin (plant, bacterial, animal), mode of action (scavengers, emitters, antioxidants, antimicrobial agents, antibiofilm agents, etc.), structure (organic, inorganic, metals, antimicrobial peptides, enzymes, antibiotics, etc.) and others. According to Vilela et al. [20], active agents include antioxidants (polyphenols, α-tocopherols, lignin, essential oils, plant extracts), CO_2_ emitters (sodium bicarbonate, citric acid, ferrous carbonate), O_2_ scavengers (iron, palladium, ascorbic acid, pyrogallol, gallic acid, glucose oxidase, laccase), ethylene scavengers (KMnO_4_, activated carbon, metal oxides, titanium dioxide) and antimicrobial agents (metals, nisin, essential oils, lysozyme, chitosan, nisin, lactoferrin).

Below, the main group of antimicrobial agents are described together with their application in the food packaging industry. First, inorganic agents, i.e., metals and metal oxides, are described. Then, various groups of organic antimicrobial agents, such as organic acids, antimicrobial peptides, enzymes, and natural antimicrobial agents of plant origin (like polyphenols, essential oils components and complex plant extracts) are described. Additionally, at the final part of the manuscript, other agents that are used in the packaging industry, but cannot be classified into other groups, are described.

### 3.1. Metal-Based Nanoparticles as Antimicrobial Agents

A new generation of packaging materials are nanocomposite films which combine the bio-based polymer (acts as a matrix) with fillers (e.g., antimicrobial agents) that are dispersed therein to improve the functional properties of material. Bio-based polymers can be, as mentioned earlier, both natural (various polysaccharides, proteins, lignins, lipids, etc.) and synthetic (natural-based or bio-based synthetic polymers like polylactic acid/PLA/and partially bio-based polymers such as polyethylene/PE/, poly(ethylene terephtalate)/PET/and polyamide/PA/)) [21], while as nanofillers metal or metal oxides are often utilized.

Metals and metal oxides are used as nanoparticles (NPs) due to their small size, favoring penetration into the interior of the bacterial cells. Although the exact mechanism of the toxicity of metal NPs and metal oxides NPs against bacteria is not fully understood, it has been postulated that it includes the generation of free radicals and reactive oxygen species (ROS), as well as the direct contact with bacteria surface that results in cell wall damage and the interruption of transmembrane electron transport [22,23,24]. ROS then induce oxidative stress, modulate the gene expression and cause the damage of important cell components, such as DNA and proteins, affects the mitochondria function and causes the cell membrane and cell wall disruption, leading finally to bacteria cell death [25]. The antibacterial activity of metal NPs is also dependent on the release of metal ions, especially when metal ions play the physiological role (such as Cu^2+^) [26]. The differences between cell wall structure and composition of Gram-positive and Gram-negative bacteria are important for the final antibacterial impact of metals NPs. It has been proved that Gram-positive bacteria are more resistant to the NPs than Gram-negative bacteria because they contain more peptidoglycan layers and teichoic acids. Metal NPs can therefore interact with numerous units of amino acids and carbohydrates of the peptidoglycan and can also be more effectively trapped by negatively charged peptidoglycan [24,27]. The importance of the charge of metal nanoparticles for its antimicrobial activity was demonstrated by Abbaszadegan [28]. Among different silver NPs tested, the positively charged NPs had the highest antimicrobial activity against all tested bacterial species, both Gram-positive and Gram-negative.

#### 3.1.1. Metal Nanoparticles

Among metal nanoparticles (NPs) which exert an antimicrobial effect, the most common used are NPs of copper, silver, selenium, and gold [16,21,29,30,31,32]. Metal NPs can be added to the polymer matrix. However, not all metal NPs are incorporated in food coating, as sometimes they are used alone or with combinations of other antimicrobials. For example, copper nanoparticles exert anti-biofilm activity against *E. coli*, *P. aeruginosa, Salmonella* Typhi, and *Shigella flexneri* at the concentration of 12.5 μM, while the minimum inhibitory concentration (MIC) is 25 μM [33].

In vitro studies showed that silver nanoparticles (Ag NPs), synthesized from *Corynebacterium glutamicum*, had the highest antimicrobial activity against *Klebsiella pneumoniae* (inhibition zone 18 mm in diameter) and were effective against *E. coli, S. aureus, Salmonella enterica, S. flexneri, P. aueroginosa, Bacillus subtilis* and *B. flexus* (clear zones of inhibition with 16 mm to 4 mm in diameter) [34]. Gold nanoparticles exerted strong inhibitory activity (with MIC value lower than that of antibiotics) against *E. coli* ATCC25922, *S. aureus* ATCC 29213, *Bacillus subtilis* ATCC 11774, *P. aeruginosa* ATCC27853 and *Candida albicans* [35]. Chitosan nanoparticles and chitosan nanoparticles loaded with Fe^2+^ or Fe^3+^, showed significantly higher antimicrobial activity (against *E. coli, S. aureus* and *C. albicans*) than chitosan and related metal ions, and Fe^2+^ ions were more effective than Fe^3+^ [36]. Table 1 presents examples of food packaging with the addition of metal nanoparticles which gave or increased the antimicrobial properties of the packaging.

#### 3.1.2. Metal Oxide Nanoparticles

Additionally, metal oxide nanoparticles can be used as nanofillers. The most common to be used in this way include zinc oxide (ZnO), titanium oxide (TiO_2_), silica (SiO_2_), aluminum oxide (Al_2_O_3_), iron oxide (Fe_2_O_3_), and copper oxide (CuO) (Table 2).

Cerium (Ce)-doped SnO_2_ nanoparticles were obtained by mixing SnCl_2_∙2H_2_O and CeCl_3_ solutions with hydrogen peroxide [47]; they were proven to show antimicrobial activity against *E. coli*. SnO_2_ nanoparticles showed inhibitory activity against *E*. *coli* ATCC 25922 and—to a lesser extent—against *S. aureus* ATCC 29213 in MilliQ water [48]. In other studies, it was proven that the addition of the ZnO nanoparticles resulted in the emergence of antibacterial activities that were not found in the pure PLA film [49]. The most potent was biocomposite PLA with 5% ZnO which reduced the *E. coli* concentration by 99.99% already after 24 h, while the 1% ZnO and 3% ZnO reached the same antimicrobial activity after 5 days. According to Sirelkhatim et al. [25], mechanisms of toxicity of NPs and their ions (e.g., silver and zinc) against bacteria include the ROS generation, with the subsequent induction of oxidative stress and irreversible damage of cellular structures resulting in bacterial death.

Although NiO nanoparticles were proven to exert antimicrobial activity against *E. coli* and *S. aureus* growth [50], and the use of chitosan-based film with the incorporation of NiO NPs had improved thermal stability and crystallinity and exhibited good antibacterial activity against *S. aureus* and *S. typhimurium* [51], NiO NPs are not often used in food packaging. This is caused by the possible negative impact of nickel on the health of consumers, especially due to the growing frequency of nickel allergies [52].

### 3.2. Organic Acids

Many organic acids have a long history as natural or traditional food additives because they decrease pH in solutions, inhibiting the growth of undesired microorganisms. One of the best known examples of organic acids used for food preservation is lactic acid, which is produced by the lactic acid bacteria during fermentation of plant raw materials (silage, pickled vegetables and fruits) or milk.

Organic acids exist in two forms—dissociated and undissociated. It has been proved that the efficiency of an organic acid in inhibiting the microbial growth depends on its pKa value, which describes the pH value at which the acid has available 50% in its dissociated and 50% in its undissociated form. Some studies revealed that only undissociated form of organic acid can pass through the walls of microorganism and alter their metabolism or exert an inhibitory effect [68,69]. It is believed that weak organic acids, such as lactate or acetate, can permeate the lipid bilayer and release their protons inside the cell, causing a drop in the intracellular pH. This causes that bacteria try to eliminate excessive protons from their cells. As protons can be effluxed from the bacteria using special proton pumps that require energy (ATP), bacteria consume energy for pH stabilization instead of cell growth [70,71]. Furthermore, the impaired transport across membranes and increased membrane permeability can even result in cell death. This means that the organic acids are good candidates to be antimicrobial agents.

Many organic acids, e.g., butyric acid, valeric acid, monopropionin, monobutyrin, monovalerin, monolaurin, sodium formate, were proved to exert antimicrobial activity [72]; however, only several of them are used in food packaging. Salts of some organic acids, such as sodium lactate, potassium lactate as well as sodium citrate, have been proved to reduce the growth of *L. monocytogens* in beef frankfurters [73]. The effectiveness of antimicrobial activity against *L. monocytogenes* can be enhanced by using a UVC treatment of the surface of frankfurters that contained potassium lactate and sodium diacetate [74].

The microbiological stability of fresh salmon slices, preserved by dipping in 2.5% (*w*/*v*) aqueous solution of sodium lactate (NaL), sodium citrate (NaC) and sodium acetate (NaA) and then stored at 1 °C was examined by Sallam [75]. All tested salts inhibited the proliferation of the food spoilage microorganisms (*Pseudomonas* spp., H_2_S-producing bacteria, lactic acid bacteria, and Enterobacteriaceae), with the order of their efficacy being NaA > NaL > NaC. The results were similar to those obtained for the rainbow trout fillets dipped for 10 min in the same solutions (2.5% NaA, NaL, and NaC) before storage at 4 °C [76]. The shelf life of the treated fillets was extended by 3 days, as all tested salts prevented the proliferation of total mesophilic bacteria, psychrotrophs, and coliform bacteria. Additionally, in this experiment, the order of antibacterial activity of organic salts used was NaA > NaL > NaC.

Salts of benzoic acid, sorbic acid and propionic acid have been used as food preservatives for many years. Although they have GRAS status, recently some of them began to raise doubts about its safety for consumer health. For example, propionic acid acts mainly against molds and can be added to food products (calcium propionate and sodium propionate are used as preservatives in bread and bakery products, grains, meat preparations, processed meat and processed fish, and animal feed) or is produced in situ during fermentation (e.g., in cheese) [77]. Recently, it was demonstrated that oral consumption of propionic acid can lead to inappropriate activation of the insulin counterregulatory hormonal network [78]. Therefore, propionate has rarely been investigated as an additive to active packaging.

Sodium benzoate and potassium sorbate are well known antimicrobial agents, and in 1995 they were reported to suppress the growth of *Zygosaccharomyces bailli* in mayonnaise [79]. Active ethylene vinyl alcohol copolymer (EVOH) film, containing the addition of 3% (*w/w*) sorbic acid–chitosan microcapsules (S-MPs), was applied to fish fillets and had stronger inhibitory capacity against *Salmonella* Enteritidis and *E. coli* than against *L. monocytogenes* [80]. The combination of sodium nitrite and sodium benzoate exerted inhibitory activity on *E. coli, S. aureus, Bacillus mucoides* and *Candida albicans,* while the combination of potassium sorbate with sodium nitrite had antimicrobial effects against *Bacillus mucoides, P. aeruginosa* and *E. coli* [81]. The fermentation of black olives in brine with the addition of 500 ppm potassium sorbate (PS) and 1000 ppm sodium benzoate (SB) caused the reduction of the yeast count, and its effect was stronger than when brine with the half lower level of PS and SB or the control brine with no preservative were used. The growth of molds was completely inhibited in all treatments during fermentation [82]. Active starch films with glycerol and potassium sorbate inhibited the growth of *Candida* spp., *Penicillium* spp., *S. aureus* and *Salmonella* spp. and extended the shelf life of refrigerated cheese by 21% [83]. Antibacterial packaging films, prepared by combining polyethylene (PE) with antibacterial agents sodium dehydroacetate and potassium sorbate, had excellent physical properties and inhibited the growth of *E. coli* more strongly than that it did the growth of *S. aureus* [84].

No *Listeria monocytogenes* were detected (8-log reduction) after 24 h of exposure to a corn zein film with lauric acid (LA) incorporated [85]. In another study, soy-based biofilms, enhanced with lauric acid (8%, wt/wt) and/or 2.5% pure nisin (4%, wt/wt), were used to protect the turkey bologna surface [86]. Films containing both LA and nisin completely eliminated *L. monocytogenes* from a 10^6^ culture to an undetectable level after 8 h of exposure at 22 °C, while films with LA alone reduced *L. monocytogenes* culture from 10^6^ to <10^2^ after 48 h.

### 3.3. Antimicrobial Peptides and Bacteriocins

Antimicrobial peptides (AMPs) are the host defence oligopeptides composed of five to over a hundred amino acids produced by various organisms, both prokaryotic and eukaryotic [87,88]. There are thousands of AMPs. The first AMPs isolated and characterized were those produced by bacteria, i.e., bacteriocins. Bacteriocins include nisin, leucocin, lactocin, enterocin, lactococcin A, bifidin, pediocin, while AMPs of other origin include, among others, cathelicidins, defensins, pleurocidin, LL-37, plectasin, protegrins, cecropins and magainins [89,90,91,92].

Most of the known and studied AMPs are cationic (positively charged) and amphiphilic (hydrophilic and hydrophobic) α-helical peptide molecules, but β-sheet structures can also be present [4]. AMPs are characterized by a broad spectrum of targeted organisms, protecting against bacteria, fungi, parasites as well as viruses. They act with high efficacy at very low concentrations and perform synergistic action with antibiotics. Due to their highly cationic character, AMPs can bind with the negatively charged bacterial cell membranes, leading to the change of their electrochemical potential. This results in cell membrane damage and the leakage of larger molecules such as proteins, destroying cell morphology and membranes, and eventually in cell death [88]. For example, cathelicidins are active against both Gram-negative (*E. coli, Salmonella* Typhimurium, *P. aeruginosa),* and Gram-positive bacteria (*S. aureus, L. monocytogenes, B. subtilis)*. They bind to lipopolysaccharide in the bacterial cell wall [93]. The antibacterial activity of magainin I molecules which were anchored to the gold surfaces was tested against three Gram-positive bacteria (*Listeria ivanovii, Enterococcus faecalis and S. aureus*), and the results revealed that the adsorbed magainin I reduced the adhesion of bacteria to the surface by more than 50%, together with killing the bacteria that nonetheless adhered to the surface [94].

Despite their name, antimicrobial peptides not only have a bactericidal effect, but they also inhibit the formation of biofilm, modulate the immune system, and have anti-cancer or anti-viral properties [89,95,96].

AMPs are quite commonly used in the food packaging industry (Table 3). Depending on the type of food and the structure of the AMPs, various strategies can be used for this purpose. For example, AMPs can be directly incorporated into the packaging polymer, can be coated onto the polymeric surface, or can be immobilized in the polymer [96]. AMPs can be stabilized by nanoencapsulation using various biopolymers and then can be delivered to food via head space or via the direct contact of active packaging system with food [92].

### 3.4. Natural Antimicrobial Agents of Plant Origin

It is estimated that there are more than 1300 plants that contain known components with antibacterial activity against spoilage microorganism and foodborne pathogens. Plant raw material is very rich in antimicrobial agents because plants naturally synthetize a wide range of compounds that protect their tissue against microorganisms and other predators [119]. Therefore, they are used as plant extracts, but also as individual fractions (purified by various extraction processes) containing mixtures of biologically active ingredients (e.g., essential oils) or as a single compound (e.g., specific polyphenolic compound). All these compounds are usually also strong antioxidants, therefore there is a great demand for them in food industry.

Antioxidants in food products play a dual role. First of all, they preserve food, protecting its ingredients from oxidation and, consequently, from spoilage and changes in flavor and aroma. However, they might also have a microbiological effect because they inactivate or inhibit the growth of microorganisms that cause spoilage of the product, as well as pathogenic microorganisms.

Antioxidants can be enzymatic and non-enzymatic. One of their most important activities includes the scavenging of free radicals (such as, among others, hydroxyl radical ^•^OH and the superoxide anion radical O_2_^−•^) by donating a single electron (SET) or by hydrogen atom transfer (HAT) [120,121,122,123]. Among strong free radical scavengers are mainly polyphenols and some antioxidant vitamins. Carotenoids are antioxidants known for their ability to quench the reactive oxygen species (ROS), mainly ^1^O_2_ [124,125]. Antioxidants include also the enzymes that catalyze the reactions of radical and/or ROS degradation. Superoxide dismutase (SOD) catalyzes the dismutation of O_2_^−•^, catalase is responsible for conversion of H_2_O_2_ into H_2_O and O_2_, while peroxidases catalyze the reduction of H_2_O_2_, organic hydroperoxides and peroxynitrite [126,127]. The antioxidant functions of enzymes also include the regeneration of glutathione (GSH) from the oxidized form (GSSG) by GSSG reductase [128]. Vitamin C, apart from its ability to donate a hydrogen atom or electron, can regenerate other antioxidants, such as α-tocopherol [129,130,131]. Besides, the activity of some antioxidants consists of preventing the ROS production, mainly by the chelation of metal ions (like Fe and Cu), inhibiting in this way the Fenton and Haber-Weiss reactions [130].

The earliest documented and most important examples of plant materials whose dual effects have been used in industrial practice are essential oils. Essential oils (EOs) are the secondary metabolites of plants. EOs are a mixture of various volatile organic compounds, such as ketones (e.g., camphor, carvone), aldehydes (e.g., citral, citronellal, cinnamic aldehyde), phenols (e.g., thymol, eugenol, carvacrol, myristicin), esters (e.g., geraniol acetate, cedryl acetate), ethers or oxides (e.g., eucalyptol, linalool oxide), terpenes (e.g., cymene, limonene, myrcene), monoterpene and sesquiterpene alcohols (e.g., linalool, citronellol, geraniol, menthol, nerol, bisabolol), and many other substances [132]. For industrial applications, EOs are extracted mainly from herbs and spices (rosemary, cloves, horseradish, mustard, cinnamon, cardamom, anis, sage, onion, garlic, parsley, oregano, basil, marjoram, savory, thyme, lavender, turmeric, mint, ginger), but also from flowers (jasmine, rose, violet), and citrus fruit [133,134,135,136,137,138,139]. Due to their antibacterial, antifungal, anti-inflammatory, anesthetic, insecticidal, and antiviral properties, EOs have many applications in medicine, but also in the cosmetic, perfumery, pharmaceutic, beverage, feed, and food industries.

The use of herbs and spices, as well as EOs extracted from them, for food preservation is aimed at extending food storage time and shelf life, reducing the number of or causing the complete elimination of pathogens and improving the overall quality of food products [133,138,140,141,142,143]. Spices and herbs are commonly added to various types of food products. They can be found in meat and meat products, fish, fruit and vegetable processing products, but also in cheese, milk, yoghurts and butter, where they inhibit pathogens (e.g., *S. aureus, E. coli* O157:H7, *C. perfringens, L. monocytogenes, Salmonella* sp., *Shigella* sp., *Pseudomonas* sp., *Vibrio parahaemolyticus*) and other microorganisms, causing food spoilage and the significant financial losses associated with it (e.g., *Botrytis cinerea, Alicyclobacillus acidoterrestris*, molds) [134,135,137,138,143,144,145].

In addition to essential oils, other ingredients of plant origin with antimicrobial potential may also be used in food technology, including polyphenols, terpenoids, alkaloids, saponins, polyamines, isothiocyanates, thiosulfinates and glucosinolates [119]. It should be highlighted that some polyphenols and terpenoids are also components of essential oils. Moreover, the biosynthesis pathways of some of EOs components have common parts with polyphenols biosynthesis; for example, some stages of the eugenol biosynthesis are the same as those for *p*-coumaric acid, caffeic acid and ferulic acid [146].

In the food industry, particular attention is paid to polyphenols, which are usually divided into 4 or 5 classes: phenolic acids, flavonoids (these include flavonols, flavanones, flavanols, flavones, anthocyanins and isoflavones), stilbenes, lignans, and sometimes coumarins [147,148]. Thanks to their structure, polyphenols are ideally suited to act as antioxidants so they can inhibit the formation of free radicals, quench already formed free radicals, inhibit initiation and interrupt initiated radical reactions [149,150,151]. In this way, polyphenols effectively inhibit lipid peroxidation, the oxidation of proteins and sugars, and oxidative damage to nucleic acids. Polyphenolic compounds contain numerous hydroxyl groups, which have an inhibitory effect because they react with the bacterial cell wall, damage the structure of cell membranes and cause the leakage of cell components. Furthermore, as antioxidants, they inhibit the formation and action of reactive oxygen species (ROS) and scavenge free radicals, and consequently reduce the oxidation–reduction potential of the environment or growth medium [139]. Thanks to this, in addition to antioxidant activity, polyphenols have a bactericidal or bacteriostatic effect against undesirable microorganisms. Among the most important mechanism of the antibacterial action of polyphenols are: reactions with proteins; inhibition of nucleic acid synthesis by bacterial cells; DNA damage; interaction with the bacterial cell wall or inhibition of cell wall formation; alteration of cytoplasmic membrane function; inhibition of energy metabolism; changes in cell attachment and inhibition of biofilm formation; and substrate and metal deprivation [148].

Many ingredients of essential oils have already been recognized as safe in the European Union (e.g., carvacrol, carvone, cinnamaldehyde, citral, eugenol, *p*-cymene, limonene, menthol, thymol) and new sources of compounds are still being sought with antioxidant and/or antimicrobial activity. Still, not all discovered substances, the addition of which to food could bring measurable benefits, have been checked for their impact on the human body, for example in terms of toxicity and allergenicity. The values of MIC have not been assessed for these characteristics, and possible synergistic or antagonistic reactions have not been assessed with other food ingredients. The necessity of such studies is confirmed by the results obtained by various authors [152,153,154].

As in the case of antioxidant potential, the location of hydroxyl groups is also of great importance for antimicrobial properties. Thymol (2-isopropyl-5-methylphenol) and carvacrol (5-isopropyl-2-methylphenol) are isomers; they have a similar molecular structure, but the –OH groups in thymol are located in the *meta* position, and in carvacrol in the *ortho* position. Dorman and Deans [155] showed that this has a significant impact on the effectiveness of action against Gram-positive and Gram-negative bacteria, with thymol having a stronger inhibitory effect. The presence of the –OH group at the C5 carbon atom is also important for the activity of flavanones and flavones against MRSA (methicillin-resistant strains of *S. aureus)* [139]. Esterification of the hydroxyl group (e.g., in carvacrol methyl ester) causes changes in the delocalized electron cloud. Such a compound loses its ability to inhibit the growth of *Bacillus cereus* [124], and its activity against other bacteria is negligible compared to that of carvacrol [155]. 

The co-occurrence of –OH groups and the electron delocalization system (in the form of double bonds in the ring) is important here. The lack of this second “element” of the molecular structure, as in menthol molecule, prevents the detachment of the proton from the –OH group and makes the antimicrobial activity much lower (Figure 1). Additionally, ring rupture weakens the antibacterial potential, which is why geraniol and nerol are much weaker inhibitors of bacterial growth than carvacrol or thymol [139,155].

Polyphenols can be used individually or in combination (mixture or as an extract ingredient), they act on both Gram-positive and Gram-negative bacteria, and their effectiveness depends, among other factors, on the dose, pH, temperature and oxygen availability [133,136,137,140,143]. For example, xanthohumol, the main flavonoid in hops, exerted high antimicrobial activity against *Bacteroides fragilis, Clostridium perfringens* and *C. difficile* [157]. Xanthohumol, naringenin, chalconaringenin and 4-hydroxy-4′-methoxychalcone inhibited the growth of *S. aureus* [158], and the presence of at least one –OH group, especially at the C-4 position, was crucial for that activity. Inactivation of the compounds might be achieved by the replacement of hydroxyl group by a halogen atom, nitro group, ethoxy group, or aliphatic groups. Hydroxycinnamic acids induced greater potassium and phosphate leakage than hydroxybenzoic acids across the membranes of *Oenococcus oeni* and *Lactobacillus hilgardii* [159]. Caffeic acid had a higher antimicrobial activity than *p*-coumaric acid due to the additional –OH group attached to the phenolic ring in the molecule [160].

Flavonoids can occur in two forms: free as “aglycons” or in the form of “glycosides”, and it has been postulated that glycosylation enhances antimicrobial activity, in contrast to their antioxidant, anti-inflammatory, anticancer and cardioprotective properties [161]. However, it has been demonstrated that flavonoid aglycones, but not their glycosides, may inhibit the growth of some intestinal bacteria [162]. The study in question revealed that quercetin 3-*O*-rutinoside had no inhibitory influence on the intestinal bacteria analyzed, and even slightly stimulated *Lactobacillus* spp. growth, whereas its aglycone quercetin exerted a dose-depended inhibitory effect on intestinal bacteria *Ruminococcus gauvreauii*, *Bacteroides galacturonicus* and *Lactobacillus* spp. The same was true for flavanones; glycosides naringin and hesperidin (flavanone 7-*O*-glycosides) had no impact, but their aglycones (naringenin and hesperetin, respectively) inhibited the growth of almost all the bacteria analyzed. The fact that polyphenols do not only act selectively against undesirable microorganisms should be considered when food packaging is designed.

Some properties of EOs, such as their volatility, low solubility in water, and sensitivity to oxidation limit their applications in producing food packaging films. Therefore, various methods were developed, such as encapsulation or incorporation into polymer matrix, to eliminate these disadvantages.

Table 4 presents the examples of active packaging in which antimicrobial agents of plant origin (polyphenols, essential oils, plant extracts) were used for food applications, together with targeted microorganisms. Although the antibacterial activity was reported in hundreds of plant species [163], extracts of only a few of them have been used in the design of active packaging. Plant extracts are complex mixtures that contain a varied range of components, such as terpenoids, phenolic compounds, alkaloids, glucosinolates, and many others, so synergistic and antagonistic interactions between the ingredients can be present. Therefore, plant extracts together with polyphenols and essential oils were presented below as a large group of antimicrobial agents of plant origin used for food preservation.

### 3.5. Enzymes

Enzymes are increasingly used as antimicrobial agents in food due to their ability to inhibit bacterial biofilm formation, either direct attacking the microorganisms and/or undergoing catalyzing reactions that result in formation of antibacterial compounds. There are three main groups of enzymes with antimicrobial and antibiofilm potential (Figure 2) [215].

As was presented by Alves [4], enzymes can act against biofilm using various mechanisms, like the degradation of structures involved in the cell adhesion and attachment to the surfaces, by the inhibition of biofilm formation, by the detachment of established biofilm or by increasing its susceptibility to antimicrobials. Among the enzymes with well-known anti-biofilm potential are lysozyme (responsible for peptidoglycan hydrolysis), amylase dispersin B (hydrolysis of poly-N-acetylglucosamine), and alginate lyase, being the most common used, followed by lysostaphin, proteinase K, and DNAse I.

Lysozyme is a natural antimicrobial compound that can hydrolyze β-(1,4) glycosidic linkage between N-acetylmuramic acid and N-acetylglucosamine in the peptidoglycan, which is the main component of the bacterial cell walls. Therefore, lysozyme can be used to inhibit bacterial growth in various food products, especially Gram-positive spoilage bacteria which have more peptidoglycan [216]. Lysozyme is used to inhibit bacterial growth in unpasteurized beer [217], meat [218], sausage [219], as well as in dairy products. Hen egg white lysozyme (300 mg/L) was tested for antibacterial activity against lactic acid beer spoilage bacteria: *Pediococcus inopinatus, Lactobacillus brevis, L. brevisimilis* and *L. lindneri* [220]. It was revealed that *P. inopinatus* was the most sensitive, while the most resistant was *L. lindneri*.

Lysozyme is the enzyme most commonly utilized in food packaging. Coatings with lysozyme extended the shelf life of mozzarella cheese [221], ‘coalho’ cheese [222], halloumi cheese [223] or gouda cheese [224], reducing the microorganism level. Edible films, based on chitosan and sodium alginate with lysozyme incorporated, had higher inhibitory activity against the fish spoilage bacteria *Pseudomonas fluorescens* and *Shewanella putrefaciens* than the monolayer film [225]. A combination of chito-oligosaccharides (COS) and lysozyme was effective against bacteria in the meat model system, resulting in the complete elimination of *E. coli, P. fluorescens* and *B. cereus* and a reduced load of *S. aureus.* The shelf life of minced meat containing COS–lysozyme mixture was extended up to 15 days at chilled temperatures [226]. The chitosan composite films with 60% lysozyme incorporation had enhanced inhibition efficacy against *Streptococcus faecalis* and *Escherichia coli* when compared to chitosan films [227].

Enzymes that can dissolve the established biofilm are DNAse I, proteinase K and dispersin B, but this ability depends on the type of biofilm [228,229]. Dispersin B is an enzyme that has ability to cleave the polysaccharide poly-N-acetylglucosamine, a component of the biofilm matrix produced by several Gram-positive bacteria such as *Staphylococcus epidermidis* and *S. aureus*. Pavlukhina et al. [230] produced the coatings by binding dispersin B to surface-attached polymer matrices (poly(allylamine hydrochloride), hydrogel matrices and poly(methacrylic acid)), proving that they inhibited biofilm formation (at least 98% reduction) by two strains of *S. epidermidis*.

The extracellular DNA plays an important role in the formation and aggregation of bacterial biofilm [231,232], and so DNAse I can be used as a biofilm inhibitor. Swartjes et al. [233] demonstrated that the inclusion of DNAse I in surface coating was effective in enzymatic cleavage and disrupting the extracellular polymeric substances produced by bacteria, and yielded a >90% reduction in adhering *Pseudomonas aeruginosa* and *Staphylococcus aureus*. Kim et al. [234] demonstrated that DNase I significantly inhibited the biofilm formation by *Campylobacter jejuni* and *C. coli* when isolated from commercially bought raw chickens.

Extracts of *Raphanus sativus* var. *longipinnatus* (Brassicaceae plant) combined with proteinase K have been shown to exert anti-biofilm activity and reduce biofilm formation by *E. coli* O157:H7 on stainless steel surfaces [235]. Cellulase and proteinase K were able to degrade *Salmonella* Enteritidis biofilms, while proteinase K mixed with chlorine had the potential to effectively control the *Salmonella* biofilms formed on the food contact surface [236]. The combined enzymes of lipase, cellulase and proteinase K inhibited the biofilm formation by *Vibrio parahaemolyticus* on different carriers, with the highest inhibition rate of 59% on nonrust steel plate [237].

Lysostaphin (Lys) catalyzes the hydrolysis of cross-linking pentaglycine bridges of the cell wall of staphylococci and so it seems to be a good candidate for a antimicrobial agent. Unfortunately, only several studies have addressed food preservation by this enzyme application. A lysostaphin-producing strain of *Lactobacillus curvatus* (Lys+) was used as starter culture in fermenting sausages contaminated with *S. aureus* and *S. carnosus* [238]. The residual activity of Lys after fermentation was sufficiently high to reduce staphylococci level by 10^4^ to 10^5^ CFU/g within 2 to 3 days to below the level of detection. Milk from transgenic cows secreting over 3 μg Lys/mL of milk was pasteurized and then processed into cheese. Although the quantity and activity of the lysostaphin decreased during cheesemaking, its level was sufficient to protect against staphylococci in resulting dairy foods [239].

### 3.6. Lactoferrin

Milk is the first nutritious product consumed by mammals, and therefore it plays a very important role in the development of the young organism. It owes its valuable properties to the right proportions of nutrients, but also to the wealth of compounds that modulate the immune system. Among the numerous protein components of milk, it is certainly worth to mention alpha-lactalbumin, beta-lactoglobulin, bovine serum albumin, immunoglobulins, glycomacropeptides, and minor proteins, i.e., lysozyme, lactoperoxidase, and lactoferrin [240]. The antimicrobial properties of milk are related mainly to lactoperoxidase, lysozyme, lactoferrin, and immunoglobulins.

Lactoferrin is an iron-binding protein in milk. It therefore has antibacterial properties, especially against bacteria requiring high iron in the environment (like coliforms) [241]. In food packaging industry lactoferrin was used in combination with various polymers. For example, lactoferrin-gellan nanoparticles (L:G) were mixed in various proportions to produce efficient antimicrobial coating [242]. The 9L:1G particles showed a MIC of 0.3 mg/mL against *S. aureus*, which was six times lower than that of pure lactoferrin (2 mg/mL). Moreover, fresh strawberries coated with lactoferrin nanoparticles showed the reduction in the weight loss and growth of mesophilic bacteria during storage, and lactoferrin coating in the presence of carboxymethylcellulose (improves nanoparticle adherence to the fruit surface) extended fruit shelf life up to 6 days. A polyethylene terephthalate film containing lysozyme and lactoferrin was used to coat salmon fillet samples before storage up to 4 days at 0 and 5 °C [243]. It was demonstrated that such a coating was able to decrease H_2_S-producing bacteria at longer storage times and higher temperatures (2.7 instead of 4.7 log CFU/g in the control sample). Edible chitosan film with lactoferrin with and without lysozyme significantly reduced the growth of *E. coli* O157:H7 and *L. monocytogenes* [244]. The functionalized films made from bacterial cellulose with lactoferrin were used as edible antimicrobial packaging for fresh sausage as a model of meat products [245]. Film had bactericidal activity against *E. coli* (mean reduction 69% in the films per se versus 94% in the sausages) and *S. aureus* (mean reduction 97% in the films per se versus 36% in the sausages).

### 3.7. Chitosan

Chitosan plays a double role in food packaging, being able to be used both as a matrix material (polymer) and as a antimicrobial agent. Chitosan is a nontoxic natural antimicrobial polymer, produced from chitin, that was approved by GRAS (Generally Recognized as Safe) by the United States Food and Drug Administration. Chitin is a polymer composed of β-(1,4)-linked N-acetyl-β-D-glucosamine, and is one of the most abundant natural polysaccharides (after cellulose). It can be found in the exoskeletons of crustaceans and molluscs and in insect cuticles as well as in fungal cells [15]. Chitosan is a deacylated derivative of chitin. Due to its biodegradability, biocompatibility, and low toxicity, good film-forming ability and high versatility, chitosan has been extensively investigated and is commonly used in various industries [246,247,248,249]. Chitosan-based polymeric materials can be formed into fibers, films, flakes, gels, powders, sponges, beads or even nanoparticles [15].

It is believed that the polycationic structure of chitosan is a prerequisite for its antibacterial activity. The most accepted explanation for chitosan’s antimicrobial activity is that its positively charged molecule binds to the negatively charged bacterial cell wall, leading to cell wall damage, altered membrane permeability, osmotic imbalance, and ultimately cell death [248,249,250,251]. It was also proved that chitosan can affect protein biosynthesis and membrane fluidity, causes an increase in intracellular ROS, damages the integrity of the cell surface architecture and might be involved in energy metabolism [246]. The antibacterial potential of chitosan depends on various factors, such as pH, temperature, molecular weight of chitosan, degree of acetylation, its physical state, concentration, ionic strength, positive charge density, chelating capacity, as well as environmental and microorganism characteristics [15,249]. The chitosan-sensitive microorganism include bacteria: *E. coli, S. aureus, Bacillus subtilis, B. cereus, Pseudomonas aeruginosa, P. fluorescens, Salmonella typhimurium, S. choleraesuis, Streptococcus mutans, S. sobrinus, S. sanguis, S. salivarius, Klebsiella pneumoniae, Enterobacter aerogenes, Xanthomonas campestris, Erwinia carotovora,* as well as yeast and mold (*Candida albicans, Aspergillus* sp. and many more [15,251]. In order to enhance the antimicrobial activity of chitosan, various modifications of its molecule were conducted to obtain chitosan derivatives with higher antimicrobial properties and better water solubility [248,252]. Chitosan has been proved to also exert antifungal activity; it is able to inhibit spore germination [253], can change the expression of genes, and can alter the activity of enzymes responsible for the hyphae growth as well as intracellular ROS level [254]. The minimal inhibitory concentration of chitosan differs depending on the microorganism tested, the chitosan form or its derivatization. Values obtained in various experiments range from 10 ppm (*Botrytis cinerea*) or 20 ppm (*E. coli, S. aureus*) to >2000 ppm for *Aspergillus fumigatus* and lactobacilli; however, the MIC values can significantly differ, even for the same tested species if experimental parameters vary [250]. For example, chitosan in a solution form and chitosan nanoparticles were tested against *Candida albicans*, *Fusarium solani* and *Aspergillus niger* [255]. It has been revealed that nanoparticles of chitosan had stronger inhibitory activity against *C. albicans* and *F. solani* than the solution form (MIC values were from 3 to up to 12 times lower, depending on the microorganism and the molecular weight of the nanoparticles). Fungal (from champignon) chitosan edible coatings caused a total growth inhibition of *Saccharomyces cerevisiae* and *Escherichia coli* at concentrations of 1% and 2%, respectively. When edible coatings were applied on fresh-cut melons, the microbial counts were reduced (up to 4 log CFU/g) [256].

It is worth mentioning that chitosan nanoparticles seem to be efficient antibacterial agent, even against pathogens with antibiotic resistance. The nanoparticles of chitosan exerted antibacterial activity against 13 WHO *Neisseria gonorrhoeae* reference strains, including those resistant to multiple antibiotics, with a minimum inhibitory concentration (MIC_90_) of 0.16 to 0.31 mg/mL and a minimum bactericidal concentration (MBC) of 0.31 to 0.61 mg/mL [257]. Chitosan was also highly effective in inhibiting methicillin-sensitive *S. aureus* (MSSA) and methicillin-resistant *S. aureus* (MRSA) strains, both in planktonic form and sessile settings, reaching biofilm inhibition percentages as high as 90% for MRSA [258]. Chitosan, extracted from *Portunus pelagicus*, had antibiofilm activity against MRSA [259]. When edible chitosan membranes were applied to whole cuts of beef and mutton, a decrease in both methicillin-resistant *S. aureus* (x = −1.95 log_10_CFU/g) and *L. monocytogenes* (x = −1.07 log_10_CFU/g) counts were observed [260]. Fatty acids, coated by chitosan nanoparticles or glycol-chitosan nanoparticles, were proven to be as effective as antibiotics against both Gram-positive (e.g., *S. aureus, S. pneumoniae, Lactobacillus, C. difficile*) and Gram-negative (*E. coli, Shigella* spp., *Salmonella* spp.) bacteria [261]. Moreover, *Staphylocooccus* was unable to evolve resistance to ethyl dodecanoate in chitosan, a fact which was proven in a 6-month experiment. Wasp chitosan nanoparticles showed inhibitory activity against the growth of beta-lactamase- and carbapenemase-producing *Klebsiella pneumoniae*, *E. coli*, and *P. aeruginosa* [262]. With all this in mind, chitosan as well as chitosan nanoparticles can be used as antibacterial agents, with strong effectiveness even against antibiotic-resistant strains.

Examples of chitosan usage as a coating material in combination with various components are also described in other chapters of this review.

### 3.8. Allyl Isothiocyanate

Allyl isothiocyanate (AIT) is a compound present in the seeds, stem, leaves, and roots of cruciferous plants, such as mustard, wasabi, and horseradish mustard. It is characterized by a strong antimicrobial activity in its vapor form against a wide range of spoilage and pathogenic microorganisms at low concentrations. Hence, it can be used to design antimicrobial packaging with a slow release of AIT to prolong the shelf life of foods [263].

An antimicrobial sachet, containing AIT encapsulated in calcium alginate beads, was developed and the effect of released AIT vapor was assessed against *E. coli* O157:H7 on spinach leaves [264]. The number of *E. coli* O157:H7 on spinach leaves (5.6 log CFU/leaf) decreased by 1.6–2.6 log CFU/leaf at 4 °C and 2.1–5.7 log CFU/leaf at 25 °C within 5 days, and the reduction level was significantly dependent on the relative humidity. Stickers containing various concentration of AIT were used against the growth of *Penicillium nordicum* in pizza at 4 °C for 30 days [265]. All tested levels significantly delay visual fungal growth in pizza in a dose-dependent manner. The AIT label, in combination with atmospheric air present in the packaging, extended the shelf life of Danish Danbo cheese from 4½ to 13 weeks, while two AIT labels extended the shelf life from 4½ to 28 weeks [266]. When AIT labels were combined with A modified atmosphere in the packaging, the shelf life of the cheese was extended from 18 to 28 weeks.

Biodegradable composite films, prepared from polylactic acid (PLA) and sugar beet pulp, were coated with coating solutions (PLA+AIT or chitosan+AIT). The films significantly inhibited the growth of *Salmonella* Stanley during 24 h of incubation at 22 °C, while the populations of *Salmonella* in controls increased [267].

### 3.9. The Reuterin System (3-Hydroxypropionaldehyde/3-HPA/, 3-HPA Dimer, Acrolein, HPA Hydrate, and 3-Hydroxypropionic Acid)

Reuterin (3-hydroxypropionaldehyde, 3-HPA) is an intermediate formed by *Limosilactobacillus reuteri* during the coversion of glycerol into 1,3-propanediol, with interesting antimicrobial activity against bacteria, fungi or even viruses [268,269,270]. In fact, it is not one compound, but a whole set, the so-called the reuterin system, which includes 3-HPA, 3-HPA dimer, acrolein, HPA hydrate, and 3-hydroxypropionic acid (Figure 3).

Antimicrobial activity of the reuterin system is not clear, but its proposed mechanisms involve the induction of oxidative damage in cells by acrolein and 3-HPA (due to depletion of free –SH groups in proteins like glutathione) or DNA synthesis inhibition due to the competitive inhibition of ribonucleotide reductases by 3-HPA dimers [269].

Therefore, reuterin has quite large potential as an antimicrobial agent in the food industry. Purified reuterin and the reuterin system compounds have been already added to various kinds of dairy products and they were proved to inhibit the growth of pathogens or spoilage microorganisms. For example, reuterin produced by *Lactobacillus reuteri* ATCC 53,608 was an effective protectant against *Escherichia coli* DH5α, *Salmonella enterica* subsp. *enterica*, *Listeria monocytogenes*, *Staphylococcus aureus* and *Penicillium expansum* without changing the quality parameters (pH, acidity, soluble solids, colour, and rheological aspects) of the fermented milk product [272]. The antimicrobial effect of reuterin combined with diacetyl against *E. coli* O157:H7, *Salmonella* Enteritidis, *L. monocytogenes* was demonstrated in fermented milk [273]. Langa et al. [274] studied cheeses artificially contaminated with *L. monocytogenes* and *E. coli* O157:H7 and then ripened for 30 days. Both pathogens were not detected in 25 g cheese made with reuterin-producing *L. reuteri* INIA P572 and 100 mM glycerol from day 7 onwards, which means that in situ production of reuterin may be an interesting alternative to improve cheese safety instead of reuterin addition to food.

As mentioned above, reuterin can be added directly to liquid food or produced in situ by properly selected strains. However, attempts are also being made to add it to coatings or food packaging. Pectin-based edible coatings with lemon essential oils and reuterin decreased viable *Penicillium* spp. conidia counts in strawberries [275]. In meat and fish products, the reuterin system decreased the level of foodborne pathogens, e.g., *Pseudomonas* spp., *L. monocytogenes*, *Salmonella* Enteritidis, *E. coli* O157:H7 [269]. Preservation of sea bass fillets by coating with active sodium alginate film with *Limosilactobacillus reuteri* (source of reuterin) and glycerol showed good antibacterial activity and caused delayed proliferation of the main spoilage microorganisms [276]. The impact of post-chill spray of the mixtures of reuterin, microcin J25, and lactic acid on the viability of *Salmonella enterica* and total aerobes on broiler chicken carcasses were also studied [277]. The results proved that a mixture of reuterin and lactic acid, sprayed onto chilled chicken carcasses, was more efficient in the reduction of *Salmonella* spp. counts than the mixture of reuterin and microcin J25. Taking into account recent studies, reuterin in combinations with various bacteriocins (microcin J25, nisin, pediocin) seems to be a promising strategy for extending the shelf life of various food products [278,279,280,281].

### 3.10. Bacteriophages

One of the most interesting but still somewhat controversial ideas is the use of bacteriophage for the development of future packaging materials. Bacteriophage is a virus, so it must find a host through which could replicate in the environment, meaning that coatings with bacteriophage must have a physical contact with food. The whole cycle of bacteriophages includes finding the host, infecting it, replication of bacteriophage, bacteria lysis and the release of new phage particles into the medium. There are many commercially available bacteriophage products that are ‘generally recognized as safe’ and approved by European Food Safety Authority (EFSA) or the US Food and Drug Administration (FDA) [282,283].

Cellulose acetate films, incorporated with a solution of bacteriophages (BFSE16, BFSE18, PaDTA1, PaDTA9, PaDTA10 and PaDTA11), showed antimicrobial activity against *Salmonella enterica* subsp. *enterica* serovar Typhimurium ATCC 14028 [284]. The antimicrobial activity of the film was shown in both the diffusion method in a liquid medium and in the method of solid medium diffusion, respectively, when incubated at 35 °C, evidenced by the growth curve and the inhibition formed.

The commercially available anti-*Listeria* phage preparation LISTEX™P100 was used for limiting the growth of *L. monocytogenes* on ready-to-eat roast beef and cooked turkey in the presence or absence of the chemical antimicrobials potassium lactate and sodium diacetate [285]. LISTEX™P100, applied without chemical antimicrobials, was an effective inhibitor and caused the reduction of *L. monocytogenes* of 2.1 log_10_ CFU/cm^2^ and 1.7 log_10_ CFU/cm^2^ (respectively, for cooked turkey and roast beef) during incubation at 4 °C. In turkey samples cooked and stored at 10 °C, both with and without chemical antimicrobials, the phage-treated samples had significantly lower numbers of *L. monocytogenes* when compared to the untreated controls throughout the 28-day storage period. In other studies, bacteriophages and their enzymes endolysins were used for the growth control of *L. monocytogenes, E. coli* O157:H7, *Shigella sonei, P. aeruginosa, S. aureus, S. epidermidis, Campylobacter jejuni, Bacillus cereus,* and *Salmonella* spp. [282,283,286].

## 4. Challenges and Possible Directions of Development in Food Packaging Industry

Food packaging is becoming more and more important. By protecting the product, it supports the producer’s interests, while by maintaining the quality and safety of food it also protects the health of the consumer. On the other hand, the impact of packaging on the environment has also become important. Apart from the aspect of reuse or biodegradability, food packaging allows producers to extend the shelf life, thus preventing food waste or the throwing away of still-good products. Taking into account that food is not always kept in the right conditions (e.g., cold chain, exposure to light) or that packaging can be damaged or unsealed, it may happen that, although the shelf life has not been reached, the product is unsuitable for consumption (it will spoil or harmful metabolites and pathogens develop). In the authors’ opinion, food producers should strive to create such innovative food packaging that will actively support the extension of the shelf life, but at the same time will provide more and more accurate information about the condition of the product. Perhaps in the future, instead of a specific date on the food packaging, there will be an inscription “fit for consumption as long as the indicator is blue” or “do not consume if the indicator turns red”, etc. Additionally, ideally, monitoring and signalling would not be conducted in a zero–one way, but the colour or other scale used for the marking will be more complex (multi-level). In such a way, the kinetics of changes could also be observed, including, for example, that the expiration day is approaching, after which the consumption of a given product will be risky or even dangerous.

The direction of further development will certainly be the search for new antimicrobial agents that can be effectively used to design active packaging, and more precisely, to find out which of the currently known ingredients are suitable for this purpose. The sheer number of plant species showing antimicrobial activity [163] prompts the search in this area. Examples of interesting plants include: *Haplophyllum tuberculatum* [287], *Caryophyllus aromaticus* and *Syzygyum joabolanum* [288], *Phytolacca dodecandra fruits, Rumex nepalensis leaves, Grewia ferruginea* [289], as well as *Sambucus nigra, Murraya koenigii, Murraya koenigii* or *Mikania glomera,* which are characterized by low MIC values against bacteria [163].

It is also worth paying attention to one more issue. Some foodborne pathogens (*E. coli, S. aureus, C. perfringens*) can enter the bloodstream, causing bacteraemia (the presence of microorganisms in bloodstream) or sepsis (defined as “a life-threatening organ dysfunction caused by dysregulated host response to infection” [290]). Depending on the methodology used, between 4.6% (explicit sepsis codes) and 33.2% (Angus sepsis codes) of hospitalizations were sepsis related to a foodborne pathogen [291]. Among the main causes of hospitalizations, the authors listed: *Clostridium perfringens, Vibrio vulnificus, Vibrio parahaemolyticus, V. cholerae*, *Listeria monocytogenes*, *Cyclospora cayetanensis* and noroviruses, but important were also *E. coli* (STEC, diarrheagenic, enterotoxigenic), *Streptococcus* spp. group A, *Salmonella typhi* and non-typhoidal, *Campylobacter* sp., *Shigella* spp., and *Yersinia enterocolitica*. It is known that *T. gondii* can be acquired by humans in several ways, such as the ingestion of contaminated food, drinking water containing the oocyst, but also by contaminated blood transfusion or organ transplantation, or by transplacental transmission [1]. This means that some foodborne pathogens can also cause BAI or HAI. Considering that these infections are largely caused by the same or related species, the presence of which is also a challenge in food, substances suspected of having antimicrobial activity can be—before being used in the human body—tested in another research model, which is food packaging. In this case, it is also important that the antimicrobial agent, after being combined with the packaging matrix, maintains its ability to inhibit the growth of microorganisms and to limit the formation of biofilm, while not affecting the quality characteristics of food and its safety for the consumer. Therefore, all antimicrobial agents mentioned in this review are good candidates to be “medical” antimicrobial agents and to be applied in the future for biomaterial coating.

## Figures and Tables

**Figure 1 ijms-24-02457-f001:**
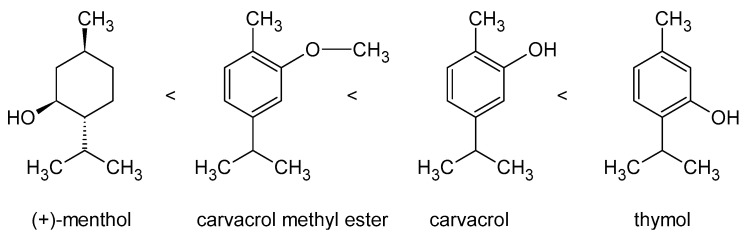
Antimicrobial activity of aromatic essential oils compounds in dependence of their structure [156].

**Figure 2 ijms-24-02457-f002:**
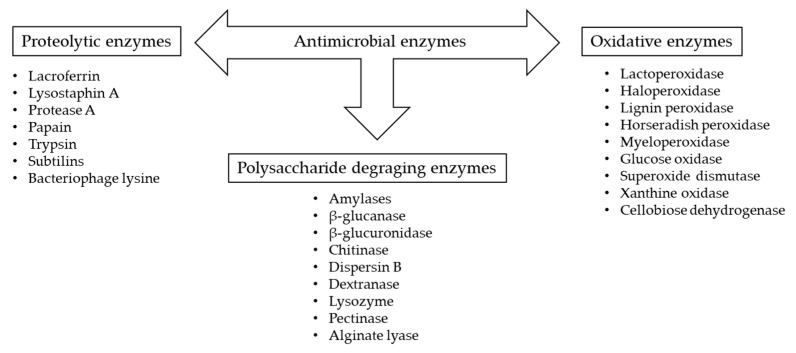
Main categories of antimicrobial enzymes based on [215].

**Figure 3 ijms-24-02457-f003:**
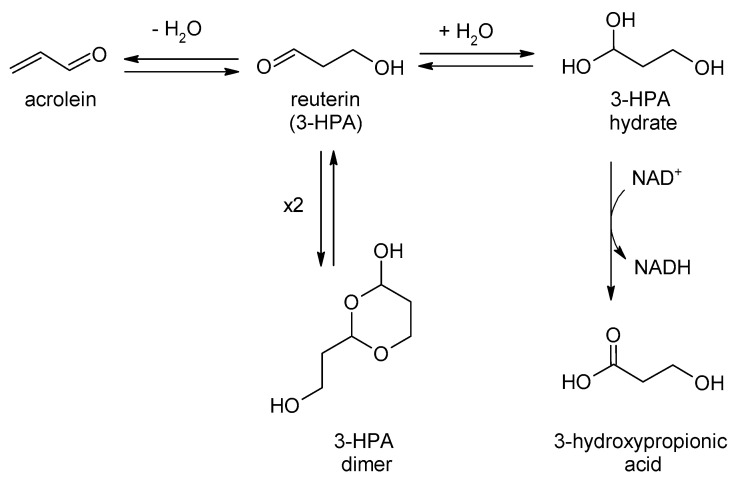
The reuterin system based on [269,271].

**Table 1 ijms-24-02457-t001:** Examples of antimicrobial metals application for food preservation.

Metal Nanoparticles	Polymer/Matrix	Characteristics	References
Ag NPs	LDPE	antimicrobial activity against mold and total bacteria in barberries packaged in Ag-LDPE film with 1 wt% AgNP	[37]
	agar hydrogel	the Ag-based packaging system was effective in inhibiting the growth of *Pseudomonas* spp. in Fior di Latte cheese	[38]
	bio-based coating with MAP	the elongation of the shelf life of Fior di Latte cheese	[39]
	polystyrene (PS) matrix	antimicrobial effect against Gram-positive and Gram-negative bacteria and yeast	[40]
	carboxymethyl CS	prepared modified antibacterial membranes could kill almost 100% of bacteria under certain conditions and inhibition zone still existed after more than 7 cycles of tests	[41]
	CS/montmorillonite nanocomposite films	CS with different molar masses and deacetylation degrees, and their modifications were used; all nanocomposite-AgNPs films inhibited the growth of *E. coli* and *Bacillus subtilis*	[42]
Ag NPs and Au NPs	CS film	good antibacterial activity against bacteria (*S. aureus, P. aeruginosa*), fungi (*Aspergillus niger*) and yeast (*Candida albicans*)	[43]
Au NPs	PVA crosslink composite films (as a crosslinking agent in film production glyoxal and/or glutaraldehyde (GA) were employed)	the improvement of banana shelf life with PVA-glyoxal-AuNPs composite film	[44]
Se NPs	multilayer laminates made of PET and LDPE	active packaging based on selenium nanoparticles prevented the oxidation of tested real food products and extended their shelf life	[45]
Se NPs	potato starch film	inhibited growth of *Salmonella* Typhimurium and *E. coli*, slightly inhibited *B. cereus*, did no inhibit *Listeria innocua*	[46]

CS—chitosan; LDPE—low-density polyethylene; PVA—polyvinyl alcohol; PHBV—poly(3-hydroxybutyrate-co-3-hydroxyvalerate); PLA—polylactic acid.

**Table 2 ijms-24-02457-t002:** Examples of antimicrobial metal oxides used in food packaging industry.

Metal Oxide Nanoparticles	Polymer/Matrix	Characteristics	References
TiO_2_ NPs	CS-based coatings	in vitro inhibitory effect on the growth of *E. coli* and *S. aureus*	[53]
	LDPE	reduction in the *E. coli* colony on fresh lettuce packed with TiO_2_ nanoparticle-coated films	[54]
	various organic and inorganic binders (e.g., PVA, polyethylene glycol, polyurethane)	bactericidal property of TiO_2_ coatings against *E. coli* O157:H7	[55]
TiO_2_ NPs and *Cymbopogon citratus* essential oil (CCEO)	CS film	treatments with CCEO and TiO_2_ extended the shelf life of minced meat as total bacterial count was in acceptable range after 10 days of storage	[56]
ZnO NPs and Ag NPs	LDPE films	the microbial load of fresh orange juice kept in packages with Ag and ZnO was below the limit of microbial shelf life (6 log CFU/mL) up to 28 days	[57]
ZnO NPs and Ag NPs and essential oils	pullulan films	ZnO NPs were active against *S. aureus, L. monocytogenes, E. coli* O157:H7, and *S*. Typhimurium, while Ag NPs were more active against *S. aureus* than *L. monocytogenes;* pullulan films containing the compounds effectively inhibited the pathogens associated with vacuum-packaged meat and poultry products stored at 4 °C for up to 3 weeks	[58]
ZnO-Ag nanocomposite	PHBV-CS	nanocomposites show great antimicrobial activity in the food packaging of poultry items	[59]
Ag-ZnO NPs	PHBV	the bionanocomposite PHBV/Ag–ZnO (10%) was the most potent against *S. aureus* and *E. coli* when compared with bionanocomposites Ag-ZnO 5%, 3% and 1%	[60]
ZnO NPs	PLA	reduction in *E. coli* growth by 3.14 log for 0.5% ZnO loading in the PLA coating layer	[61]
ZnO-SiO_2_	CS, PVA	antibacterial activity against *S. aureus* S33R and *E. coli* IRAQ 3; increased the shelf life of bread	[62]
SiO_2_	PHBV	the maximum antibacterial activity (about 94.7% growth inhibition for *E. coli* and 92% for *S. aureus*) was obtained for PHBV/SiO_2_ (2.0%)	[63]
MgO NPs	carboxymethyl-CS	CM-CS/MgO composites exhibited antimicrobial activity against *L. monocytogenes* and *Shewanella baltica*	[64]
	PLA	biofilms with 2% MgO NPs caused progressive damage and death of nearly 46% of *E. coli* bacterial culture after 12 h treatment	[65]
Aluminum-doped zinc oxide (AZO)	PLA	strong antibacterial activity against *E. coli*	[66]
Al_2_O_3_–Ag and TiO_2_–Ag composite NPs	epoxy polymer	disc diffusion assays proved antimicrobial potential against *E. coli* DH5α and *S. epidermidis* NCIMB 12,721	[67]

CS—chitosan; LDPE—low-density polyethylene; PVA—polyvinyl alcohol; PHBV—poly(3-hydroxybutyrate-co-3-hydroxyvalerate); PLA—polylactic acid.

**Table 3 ijms-24-02457-t003:** Examples of AMPs and bacteriocins application in active food packaging systems.

Peptide	Polymer/Packaging Material	Characteristics	References
Nisin	PLA	significantly inhibited growth of *L. monocytogenes* and *Salmonella* Enteritidis in liquid egg white and of *E. coli* O157:H7 in orange juice	[97]
	PLA	completely inactivation of *L. monocytogenes* growth in skim milk and liquid egg white	[98]
	PLA	reduction of *E. coli* O157:H7 count in strawberry puree at 22 °C	[99]
	Starch–halloysite nanocomposites	films with 6 g/100 g nisin completely inhibited *L. monocytogenes* on Minas frescal cheese surface	[100]
	cellulose	inhibited growth of *L. monocytogenes* and *S. aureus* in minimally processed mangoes	[101]
	cellulose	antimicrobial effect in vitro against *S. aureus* and *L. monocytogenes*	[102]
	plastic film	reduction of *L. monocytogenes* level in cold-smoked salmon samples	[103]
	edible films from whey protein isolates, soy protein isolates, egg albumen and wheat gluten	inhibitory activity against *L. monocytogenes* strain V7	[104]
Nisin and ε-polylysine	chitosan coating	decreased growth of microorganism (yeast and mold, total viable counts, total coliforms counts, *S. aureus* and *Pseudomonas* spp.) in fresh-cut carrots	[105]
Nisin with calcium propionate	edible zein film coatings	strong inhibitory impact on the growth of *L. monocytogenes* in film-coated ready-to-eat chicken	[106]
Nisin and bacteriocin-like substance (BLS) P34	liposome-encapsulated	reduced population of *L. monocytogenes* during 21 days of storage of Minas frescal cheese at 7 °C	[107]
Nisin combined with Grape Seed Extract (GSE) or Green Tea Extract (GTE)	soy protein film	the greatest inhibitory effect against *L. monocytogenes* was observed in turkey frankfurters coated with film with 1% GSE and 10,000 IU/mL nisin	[108]
Bacteriocin KU24		in vitro inhibitory effect against methicillin-resistant *S. aureus*	[109]
Enterocin 416K	LDPE film	significant decrease in *L. monocytogenes* viable counts in frankfurters	[110]
Enterocins A and B	alginate films	in combination with high-pressure processing (HPP) reduction of *L. monocytogenes* level in sliced cooked ham	[111]
Lactocin 705 and AL705	LDPE	in vitro antimicrobial activity against *Lactobacillus plantarum* CRL691 and *Listeria innocua* 7	[112]
Lacticin 3147 and nisin	cellulose-based bioactive inserts and anti-microbial polyethylene/polyamide pouches	reduced the population of lactic acid bacteria in sliced cheese and ham stored in modified atmosphere packaging (MAP) at refrigeration temperatures; nisin-adsorbed bioactive inserts reduced levels of *L. innocua* and *S. aureus*	[113]
Pediocin	PLA/sawdust particle (SP) biocomposite film	potential inhibition of*L. monocytogenes* (99% of total listerial population) on raw sliced pork during the chilled storage	[114]
Pediocins	cellulose acetate	inhibiting growth of *L. innocua* and *Salmonella* sp. on slices of ham	[115]
Plantaricin BM-1	PET/polyvinylidene chloride/retort casting polypropylene (PPR) plastic multilayer film	decreased the viable counts of *L. monocytogenes* in meat at 4 °C	[116]
Gramicidin A	multilayer biofunctionalized thin films cationic poly(L-lysine)	inhibitory activity against *Enterococcus faecalis*	[117]
Antimicrobial peptide mitochondrial-targeted peptide 1 (MTP1)	PET	reduction in total aerobic mesophilic bacteria (APC) and yeasts on ricotta cheese and meat samples	[118]

PLA—polylactic acid; CS—chitosan; LDPE—low-density polyethylene; ALG—alginate; PET—polyethylene terephthalate.

**Table 4 ijms-24-02457-t004:** Examples of antimicrobial agents of plant origin and their application for food preservation.

Antimicrobial Agent	Polymer/Matrix	Characteristics	References
**Polyphenols**			
Caffeic acid (CA)	CS	CS-CA postharvest treatment of mulberry fruit improved the quality and extend the shelf life during cold storage	[164]
Salicylic acid (SA)	CS	CS-SA coating inhibited chilling injury better than SA or CS alone and increased the antioxidant enzyme activities	[165]
Lauric acid (LA)	CS edible coating	incorporation LA increased the inhibitory effects against the spoilage bacteria growth, and LA almost completely protected the fresh beef samples against the discoloration after 21 days of storage	[166]
Ellagic acid	candelilla wax	significant reduction of the damage caused by *Colletotrichum gloesporioides* and extended the shelf life of avocado	[167]
Ferulic acid (FA)	CS films	FA-CS coating effectively extended the shelf life of refrigerated pork to 7 days; FA-CS had higher antibacterial activity than CS coatings (reduction of total viable counts by 1–2 log)	[168]
Gallic acid (GA)	CS and PE films	coating with CS-GA film improved quality of white button mushroom (*Agaricus bisporus*) in comparison to CS and PE films	[169]
Quercetin and Ag NPs	PVC-based film	PVC-based films with quercetin and AgNPs proved to be highly effective in inhibiting bacterial growth of food pathogens (*E. coli*, *S. Typhimurium* and *L. monocytogenes*)	[170]
Hydroxyapatite-Quercetin (H-Q) complexes	ALG edible coatings	H-Q alginate coatings inhibited the spoilage bacteria (*Pseudomonas* spp. and *Enterobacteriaceae*) growth and preserved the quality of chicken fillets for 11 days at 6 °C	[171]
Curcumin	PLA-based composite films	number of bacteria decreased by 1 to 2 logarithmic cycles	[172]
Curcumin	PBAT film	only curcumin-PBAT film showed a slight antibacterial activity against *E. coli* and *L. monocytogenes* (reduction of 1–2 log CFU)	[173]
Curcumin + ZnO	CMC film	only ZnO and ZnO/curcumin films had antibacterial properties	[174]
Fenugreek essential oil (FEO) + Curcumin	PLA composite film	good antibacterial and antioxidant properties of PLA-FEO-curcumin composite film	[175]
Cinnamaldehyde (CIN)	PLA and starch mono- and bilayer films	monolayer and bilayer films with CIN had bacteriostatic effects to *E. coli* and *L. inocua*	[176]
Resveratrol (RS) and its inclusion complex (IC) with hydroxypropyl-γ-cyclodextrin	film based on cellulose derivatives (hydroxyethylcellulose and cellulose acetate)	antimicrobial activity of films against *Campylobacter jejuni* and *Campylobacter coli*	[177]
Resveratrol (RS) and eugenol (EUG)	CMC films	increasing the concentration of RS or EUG in CMC films caused an increase in antimicrobial effects against *L. monocytogenes, S. aureus, Salmonella* Enteritidis, and *E. coli*—the antagonistic effects of the combined use of RES and EUG compared with their use alone	[178]
Lignin nanoparticles	PVA, CS	the inhibition of the bacterial growth of *Erwinia carotovora* subsp. *carotovora* and *Xanthomonas arboricola* pv. *pruni*	[179]
**Essential oils and their ingredients**			
Thymol and montmorillonite D43B	PLA	addition of 8 wt.% thymol and 2.5 wt.% D43B significantly increased the antibacterial activity against *E. coli* and *S. aureus* 8325-4 at all tested temperatures	[180]
Thymol (T) and eugenol (E)	PLA, poly (adipic acid succinate), and poly (succinic acid succinate) (PBS).	antibacterial activity, determined using disk diffusion method, proved that PBS/E was the most efficient against *E. coli, S. aureus, Bacillus tequilensis, B, subtilis* and *B. pumilis,* while against *Stenotrophomonas maltophilia* PLA/E was better	[181]
Thyme essential oils (TO)	β-Cyclodextrin capsule	the growth of *Alternaria alternata* was inhibited significantly by the addition and exposure to TO:β-CD as measured by both the agar dilution and the headspace method	[182]
Thymol and carvacrol	edible starch films	carvacrol and thymol presented a fungistatic effect on *C. gloeosporioides* growth on coated mango and papaya	[183]
Thyme oil (TEO) and clove oil (CEO)	PLA-PBAT film	complete killing of *S. aureus*, that is a reduction from 6.5 log CFU/mL to 0 log CFU/mL, was observed on the 10 wt% CEO incorporated composite film; CEO and TEO composite films inhibited *E. coli* biofilm by 93.43% and 82.30%, respectively	[184]
Thyme volatile oil (TVO), chitosan (and chitosan nanoparticles (CS_NP)	edible coating of CS, CS_NP and thyme volatile oil encapsulated CS_NP (TVO-CS_NP)	edible CS coating considerably extended the shelf life of basil leaves, especially TVO-CS_NPs coating (2.4-fold higher shelf life than the control)	[185]
Mediterranean propolis (EEP) and Thymus vulgaris essential oil (TV-EOs)	PLA film	EEP showed the best inhibitory effect on *S. aureus* and *Penicillium* sp. (the diameters of the inhibition zones were 12.1 mm and 11.58 mm, respectively); antimicrobial activity of the films showed that films containing 10 wt% EEP inhibited the growth of *Candida albicans* and the combination of EEP and TV-EOs in the PLA matrix showed a synergistic effect against *E. coli*	[186]
D-limonene nanoemulsion (DLN) and D-limonene essential oil (DLEO)	-	antibacterial activity was proved against *S. enterica, E. coli, S. aureus* and *L. monocytogenes*, MBC value of DLN was lower than MBC of DLEO	[187]
D-limonene	edible fish gelatin- CS film	strong antibacterial activity against *E. coli*	[188]
Limonene–liposome	ALG coating	the limonene–liposome-treated blackberries had significantly lower visible mold incidence compared to non-coated berries on days 16 and 20 days of storage, whereas the ALG-coated berries had significantly lower visible mold incidence compared to the control on the 16th day only	[189]
Monolaurin, eugenol, oregano, and thyme essential oil	zein-based films	zein films with thyme were not active against the tested microorganisms; significant inhibitory effect was observed against *S. aureus*, when oregano, monolaurin, or eugenol were added; films with eugenol were active against *E. coli, Aspergillus niger* and *Candida albicans*	[190]
Oregano oil (OO)	sorbitol-plasticized whey protein isolate (WPI) films	growth rate of total flora (total viable count) and pseudomonads were significantly reduced by a factor of two with the use of OO-films, while the growth of lactic acid bacteria was completely inhibited	[191]
Origanum vulgare essential oils (OO) and 1 or 2% of grape seed extract (GSE)	CS coatings	turkey breast meat coated with CS-based coating with GSE and OO showed reduced count of total viable count, Enterobacteriaceae, *Pseudomonas* spp., lactic acid bacteria (and yeast-mold on day 20 of cold storage	[192]
Carvacrol (CRV)	pectin/sodium alginate matrix	microencapsulated CRV and the non-encapsulated CRV treatments significantly reduced the populations of yeast, mold, *E. coli* and mesophilic aerobic bacteria	[193]
	edible coating of cassava starch	coatings with CRV reduced the counts of *E. coli* and *Salmonella* Typhimurium by ~ 5 log CFU/g; *Aeromonas hydrophila* was reduced by ~ 8 log CFU/g, and *S. aureus* was reduced by ~2 log CFU/g on the 7 day of storage of minimally processed pumpkin	[194]
	gum arabic (GA) and cCS	CRV, as well as CRV-CS or CRV-GA coatings on poultry reduced *C. jejuni* from day 0 through 7 by up to 3.0 log_10_ CFU/sample; CRV-CS coatings reduced total aerobic counts when compared with non-coated samples for a majority of the storage times	[195]
Clove essential oil (CEO)—encapsulated in mesoporous silica nanoparticles (MSN)	PLA and polycaprolactone composite films	the higher the content of MSN/CEO/PLA, the better the antibacterial effect against *E. coli* and *S. aureus*	[196]
Lemongrass oil, citronella oil and cajeput oil	edible CS or ALG film	lemongrass oil was the most effective inhibitor of *A. niger* (MIC of 10 μL/mL), citronella oil and cajeput oil MIC were of 20 μL/mL, with elongation of shelf life of mango coated by ALG film with lemon grass oil from 10 to 14 days	[197]
Lemongrass oil	ALG edible films	antimicrobial activity of ALG-based films against *E. coli* ATCC 25922 was not dependent on the type of encapsulating agent; lemongrass oil-ALG film has MIC 0.5%, while ALG film 0.6%	[198]
Encapsulated citronella oil (complex coacervation of gelatine with carboxymethylcellulose or with gum Arabic)	paper coatings	the antimicrobial activity of released citronella vapors on *E. coli* and *S. cerevisiae* (for paper coating 30 g/m^2^*)*	[199]
Lemongrass oil (LMO) microcapsules	edible coating based on sodium caseinate as wall material	films containing LMO at concentrations of 1250, 2500 and 5000 ppm were able to inhibit growth of *E. coli* ATCC 25922 and *L. monocytogenes* ISP 65–08 in liquid cultures	[200]
Citronella essential oil (CEO)	CS film with ZnO and Ag NPs	the strongest antimicrobial activity was exhibited by the CS-ZnO-AgNPs membrane, with the inhibition diameter being ~30 mm for *S. aureus* and *E. coli* and over 20 mm for *C. albicans*	[201]
Peppermint (menthol) and Origanum vulgare (carvacrol) essential oil	PE-coated films	increment of menthol and carvacrol concentration in PE film coating leads to an increase in the antimicrobial activity of films against *E. coli, S. aureus, L. innocua*, and S. *cerevisiae*	[202]
Kaffir Lime Leaf Essential Oil (*Citrus hystrix* DC)	edible coating with tapioca flour	edible coating, with the addition of kaffir lime leaves essential oils, decreased the microbial growth (total plate count) of beef sausage	[203]
Cinnamon-perilla essential oil (C-PEO)	edible composite films based on CS_NP	the shelf life of fresh red sea bream fillets wrapped in tested composite film was extended to 6–8 days	[204]
Cinnamon leaf (CLO) and garlic oils (GO)	β-cyclodextrin (β-CD) microcapsules	good antifungal activity of CLO:β-CD and GO:β-CD microcapsules against *Alternaria alternata*	[205]
**Plant extracts**			
Green tea extract (GTE)	CS film	GTE-CS film inhibited growth of *L. inocua* and *E. coli* K12 to undetectable levels in tryptic soy broth after 24 h exposure	[206]
Green tea water extracts (GTWE)	CS coating	GTWE inclusion in pork samples reduced growth of mesophilic and psychrotrophic bacteria	[207]
Extract from red grape seeds (Vitis vinifera) (GSE)	CS film	GSE-CS film caused strong inhibition of *S. aureus, E. coli, L. monocytogenes, Brochothrix thermosphacta, Acinetobacter guillouiae, Enterobacter amnigenus,* and *P. aeruginosa.*	[208]
Grape seed extract (GSE)	edible coatings and films based on CS	GSE-chitosan film inhibited growth of *L. inocua* and *E. coli* K12	[209]
Flavonoids extracted from *Moringa oleifera* seed coat	-	exhibited antibiofilm potential against *S. aureus, P. aeruginosa* and *Candida albicans*.	[210]
*Paeonia rockii* extracts (PPR) dispersed in chitosan (CS)	polysaccharide gels	coatings has antifungal activity and was able to prolong the shelf life of strawberries to about 16 days	[211]
Tomato Plant Extract (TPE)	edible CS coating	both CS and TPE alone reduced the mesophyll count of sierra fillets stored on ice during 15 days compared to control fillets, but a synergistic activity TPE-CS was the best antimicrobial treatment	[212]
Blueberry fruit and leaf extracts (BLE)	CS coatings	BLE showed good antimicrobial activity against *S. aureus, L. monocytogenes*, *Salmonella* Typhimurium and *E. coli*, with a minimum inhibition concentration from 25 to 50 g L^−1^.	[213]
Pomegranate Peel Extract (PPE)	CS_NPs	pure PPE and PPE-loaded CS_NPs effectively retarded the growth of *S. aureus* with MIC of 0.27 and 1.1 mg/mL, respectively; *E. coli* was not sensitive	[214]

CS—chitosan; PE—polyethylene; PVC—poly(vinyl chloride); ALG—alginate; PLA—polylactic acid; PBAT—poly(butylene adipate-co-terephthalate); CMC—carboxymethyl cellulose; PVA—polyvinyl alcohol; NPs—nanoparticles.

## Data Availability

Not applicable.

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
