# Peer review of "Antimicrobial Compounds in Food Packaging"

_ijms, 2023, doi:10.3390/ijms24032457_

Round 1

Reviewer 1 Report

In this review the main groups of antimicrobial agents used in food packaging are described. The manuscript topic is of high interest to preserve the food quality by using the appropriate package to resist at microbial attack.

However, I have some comments to improve  the manuscript quality.

1.  Overall, the manuscript needs to be more clearly structured. For example, in the introduction of section 3 is presented a classification of antimicrobial agents  (R 169-174) that not are found  in the next sections. The sections 3.1….3.6 are not in line with the classification from section 3.

2.      In the Table 1 some rows can be combined: i.e. nisin in PLA … can be presented in one row with indication of all references.  I suggest using one single term for cellulose: cellulose polymer or cellulose…

3.      I suggest that the Section 3.2  entitled Metals as antimicrobial agents  to be re-organized in 2 sub-sections: metals nanoparticles and metal oxides nanoparticles.

4.      The Conclusions section is not addressed to the review description.

Author Response

Our answer for the Reviewer #1 comments are in the attached file.

Reviewer 2 Report

This paper introduces antimicrobial agents that are currently commonly used in packaging for the food industry and demonstrates the promise of these agents in food packaging. However, there are still some issues to be addressed. A moderate revision is suggested before its acceptance.

1.        Detailed address including the country should be provided.

2.        The abstract is too short. Two or more sentences to show the background and aim of this review is necessary.

3.        Some of the terms used in the essay are too colloquial. It is recommended that the whole text be checked to improve the professionalism and written form of the text.

4.        There are many unclear expressions, wording errors, punctuation errors, and grammatical errors in the article, so please review the whole article again and make corrections carefully.

5.        What is the significance of the reference to smart packaging in the introduction to food packaging? Also, how does "as can prevent throwing away wholesome and safe products" in lines 139-140 extend the shelf life of food?

6.        More introduction on the different packaging materials should be provided with supporting articles: Biobased materials for food packaging; Nanomaterials 10 (1), 150, 2020; Packaging and degradability properties of polyvinyl alcohol/gelatin nanocomposite films filled water hyacinth cellulose nanocrystals; Nanomaterials 12 (18), 3158, 2022; etc. In addition, the chitosan is also an common materials used in antimicrobial for food packaging, please refer: Sources, production and commercial applications of fungal chitosan: A review

7.        The classification of organic, metallic and other active agents should be based on chemical composition, not structure.

8.        When classifying antioxidants, scavengers, etc., please give the basis for the classification.

9.        More discussion on the antioxidant and antibacterial compounds should be proposed with supporting articles: Characterization of antioxidant and antibacterial compounds from aerial parts of Haplophyllum tuberculatum; etc.

10.    What is the significance of the reference to the two forms of organic acids present in section 3.3? It is also advisable to introduce the antimicrobial mechanism of organic acids by sorting out the logic according to the degree of antimicrobial effectiveness (e.g. from energy consumption, growth inhibition to cell destruction).

11.    Do the essential oils of plants in natural antimicrobial agents contain substances such as polyphenols as mentioned below? Do components such as alkaloids and other components constitute a classification that is independent of each other?

12.    When Reuterin is mentioned first in the title of 3.6.4, a note should be made in time to explain.

13.    The conclusion section is too brief in summarizing the full antimicrobial package, please flesh it out. Also, the introduction of subsequent challenges is too redundant. Therefore, please rewrite the conclusion section.

14.    The last section should include more perspectives to provide the challenges and possible solutions for guiding the future studies to readers.

Author Response

Our answer for the Reviewer #2 comments are in the attached file. Thank you very much for your hard job!

Round 2

Reviewer 1 Report

Dear authors,

The manuscript was deeply revised and improved. I agree the response of the authors and I accept the revised version.

Reviewer 2 Report

 Authors have revised the manuscript well. Accept in present form.